# The Relationship between Changes in GRIT, Taekwondo Ability, and Academic Achievement of University Students Majoring in Science and Engineering and Participating in Taekwondo Class

**DOI:** 10.3390/ijerph18105361

**Published:** 2021-05-18

**Authors:** Ji-Hoon Lee, Jin-Hui Cho, Yi-Sub Kwak, Byoung-Goo Ko, Eun-Hyung Cho

**Affiliations:** 1Daegu Gyongbuk Institute of Science Technology (DGIST), Daegu 42988, Korea; vastlee4@dgist.ac.kr; 2Department of Nursing, Dongnam Health University, Suwon 16328, Korea; jhcho@dongnam.ac.kr; 3Department of Physical Education, Dong-Eui University, Busan 47340, Korea; ysk2003@deu.ac.kr; 4Korea Institute of Sport Science, Seoul 01784, Korea; bgko@kspo.or.kr

**Keywords:** taekwondo, GRIT, taekwondo ability, academic achievement

## Abstract

GRIT, which was conceptualized by the American psychologist Duckworth, was designed by grouping growth (G), resilience (R), intrinsic motivation (I), and tenacity (T), which means continuing to be patient and put in effort to achieve goals without being frustrated by adversity or failures experienced in the process of striving toward one’s goals. The purpose of this study was to determine GRIT changes caused by participation of students majoring in science and engineering in taekwondo class. Effects of taekwondo ability on GRIT and academic achievement were also examined to determine structural relationships among taekwondo ability, GRIT, and academic achievement. We selected a total of 305 students (204 participants and 101 non-participants) as research subjects and conducted a GRIT (preliminary) measurement. After one-year of taekwondo class, we collected and statistically processed the data of GRIT (post) measurement, taekwondo ability, and academic achievement of the participants. Reliability analysis, technical statistics, paired sample t-test, correlation analysis, and path analysis were performed. Changes in the GRIT values of the participants were found to be greater than those of non-participants. It was also found that taekwondo ability, GRIT, and academic achievement had significant correlations with each other. Finally, it was found that the higher the taekwondo ability, the higher the academic achievement and the higher the GRIT. Moreover, the higher the GRIT, the higher the academic achievement. Taekwondo training increased the GRIT values of participants. In addition, the taekwondo ability had positive effects on GRIT and academic achievement. GRIT also had a positive effect on academic achievement. Thus, there were structural relationships among taekwondo ability, GRIT, and academic achievement.

## 1. Introduction

By the time students who are majoring in science and engineering graduate from university, they are at the starting lines of their true lives, and therefore comprehensively checking for and utilizing advantages in all fields throughout university life, not only in terms of scholarships, but also in terms of psychological and emotional aspects, such as personal values, identity, and anxiety; social aspects, such as interpersonal relationships, career paths, and marriage preparation; and physical aspects such as physical strength and health care [1]. In this process, they hope to find a solution to their concerns about their abilities, their future, and their success in life. In other words, they continually raise the question of whether they can achieve their desired goals with their own abilities and succeed, and how can they improve if their abilities are insufficient, and they grow up confronting this reality. The answers to the questions they raise are naturally connected to “GRIT”.

GRIT, which refers to the core ability that an individual must possess to achieve success, is a non-cognitive predictor that is common in those who have achieved excellent results in various fields, such as management, sports, arts, and medicine [2,3]. GRIT, which was conceptualized by the American psychologist Duckworth [4], was designed by grouping growth (G), resilience (R), intrinsic motivation (I), and Tenacity (T), which means continuing to be patient and put in effort to achieve the goals without being frustrated by adversity or failures experienced in the process of striving toward one’s goals [5,6]. People with high GRIT strive for a long period of time to achieve one goal they set, while those with low GRIT are characterized by frequent changes in their goals and a low degree of continuous effort [5,6,7]. Previous studies have indicated that GRIT is a psychological trait that helps predict future achievement and success, and that significantly predicts individual achievement and performance while also affecting positive psychological factors such as happiness [8].

University students majoring in science and engineering can predict their abilities and future achievements and success based on their GRIT, and if there is a problem with their GRIT, they will have the opportunity to correct it in the course of their university life. GRIT is not innate and can be changed by learning [4,6,7]. In the short term, students can pursue changes in academic achievement through GRIT. This is also confirmed by previous studies [9,10,11,12,13,14] showing that GRIT has a significant correlation or positive effect on academic performance.

This leads to the question what kind of learning is required to improve GRIT? In his book [6], Duckworth explained the direction of growth using keywords, such as “make your interests clear”, “do qualitatively different practice”, “have a high sense of purpose”, and “willingness to try to bounce back up, have hope”. He also stated that exercise is one of the most efficient ways to achieve GRIT growth by showing that growing children achieve GRIT growth through participating in sports and physical activity. Studies have shown that people who actually participate in exercise have higher levels of GRIT strength [15] and that GRIT increases confidence in sports and intentions to participate in exercise [16], thus indicating that participation in exercise has a close interrelationship with GRIT growth [17].

Another question that comes to mind is, what kind of exercise and of what length should we do to grow our GRIT and realize our goals? In this study, we attempted to find an answer to this question by focusing on taekwondo. At present, many researchers are actively conducting studies in their respective fields of research to address this question. However, they have not provided any information on the type of exercise that can grow GRIT in the most effective manner. Only studies of tracebacks from GRIT and studies of how GRIT affects multiple social and psychological factors and competitiveness have been conducted, due to obstacles such as the long-term study period and high costs required. In particular, as studies on improving taekwondo athletes’ performance and injury are not part of the mainstream because related studies are focused on the Olympic Games, it is difficult to find information on how taekwondo affects GRIT. To overcome these constraints, we need to conduct experimental studies to analyze changes in GRIT according to taekwondo training, such as Duckworth’s GRIT research, as well as integrated studies to confirm that this GRIT, which has grown through such a process, has an impact on achievement and success.

Taekwondo is a martial arts sport that allows practitioners to learn self-realization, namely through the concept of “Do”, and apply it to their lives by cultivating their spiritual world through physical training and exercise. Unlike other sports, taekwondo is not about paying attention to participation in the sport itself, but rather about having a positive effect on the behavior and spirit of the practitioner through the training process [18]. The systematic training program and training principles of taekwondo have played an important role in making it a respected martial arts sport and an official Olympic sport that has rapidly come to be practiced by more than 80 million people in 203 countries across different races, ideologies, and religions [19]. In particular, taekwondo training is expected to go hand in hand with sand grain growth and academic performance, as taekwondo educational programs include educational content that leads to GRIT growth, such as patience, sincerity, confidence, passion, stoicism, and a never-give-up spirit [19,20]. If such taekwondo training is found to be a clear cause of GRIT growth and academic achievement, the basis for a new paradigm of taekwondo training will be formed, and it will extend beyond the constraints of existing studies and contribute to providing core information on GRIT growth and academic achievement.

Therefore, the purpose of this study is to confirm the changes in GRIT in university students majoring in science and engineering who participate in a taekwondo class, and to verify the effects of taekwondo ability on GRIT and academic achievement. In this context, this study attempted to provide information on GRIT growth as it relates to taekwondo training and to confirm the influence and structural relationship between taekwondo ability and academic achievement.

As a result, the following research hypotheses are proposed.

**Hypothesis** **1.**
*The GRIT of university students majoring in science and engineering and participating in taekwondo class will be higher than that in the same before participating in taekwondo class.*


**Hypothesis** **2.**
*Their GRIT improvement will be greater than that in those who are not participating in the class.*


**Hypothesis** **3.**
*Their taekwondo ability will affect their GRIT and academic achievement.*


**Hypothesis** **4.**
*Their GRIT will affect their academic achievement.*


## 2. Materials and Methods

### 2.1. Description of Participants and Type of Sample

Among the non-probability sampling methods, this study used the convenience sampling method to sample 320 students, including both participants and non-participants of a taekwondo class, among university students majoring in science and engineering at K and D universities in Korea. The questionnaire results of 305 participants (204 taekwondo class participants forming the experimental group, 101 non-participants forming the control group) were used for the study, after excluding data from 15 subjects who gave up the taekwondo class halfway and whose answers were judged to be unreliable. All the participants in the taekwondo class as well as non-participants were freshmen in university, and they were composed of students who lived in the dormitory during the semester and who took the same classes in the same curriculum aside from taekwondo. Of the total 204 participants in the taekwondo class, 138 were men and 66 were women; of the 101 non-participants, 57 were men and 44 were women. This study also attempted to minimize ethical issues by obtaining consent from all subjects for the collection and utilization of their personal information and the provision of personal information to third parties.

### 2.2. Instrument

#### 2.2.1. GRIT

“Grit-S” [21], which is an improved version of “Grit-0” [4] (which was developed by Duckworth et al.) to further increase reliability and validity, was used as a GRIT measurement tool. In this study, the adapted GRIT scale was revised and supplemented to match this study through a pilot test, and the validity and reliability were verified before use. The GRIT scale consisted of a total of 10 questions, and each question was responded to on a Likert scale configured with a five-step interval scale from “always (five points)” to “never (one point)”. Cronbach’s α of the GRIT scale question was measured at 0.773, and the result was derived using the total score of the 10 questions. 

#### 2.2.2. Taekwondo Ability

Students who participated in the taekwondo class at K and D universities participated in classes for two hours a week at each university for a year (over two semesters), excluding vacations. The taekwondo curriculum includes theories, attitudes, and practical skills, such as philosophy, psychology, basic skills, poomsae, and competition; this is based on the official curriculum presented by Kukkiwon [18,19]. To measure taekwondo ability, theory, attitude, and practical skill evaluations were conducted once a semester for a total of two times, and they were scored as follows: A (90 or more)—five points, B (80–89)—four points, C (70–79)—three points, D (60–69)—two points, and F (less than 60)—one point.

#### 2.2.3. Academic Achievement (GPA-Grade Point Average)

The GPAs of the year during which the K and D university students participated in the taekwondo class were used as data, and the perfect score in both universities is 4.50. Taekwondo classes are required at both universities for liberal arts degrees, and students are given grades of pass or fail, which are not reflected in the GPA.

### 2.3. Design and Procedure

This study was conducted at K and D universities in South Korea from March 2019 to February 2020, and it was conducted according to the research procedure shown in Figure 1.

### 2.4. Data Analysis

SPSS 18.0 for windows and LISREL (linear structural relationship; ver. 8.80) were used to analyze the data collected in this study through the search process, and the following specific analysis method was used: first, the paired sample *t*-test was performed to examine changes in GRIT of taekwondo class participants (experimental group) and non-participants (control group). Second, Pearson’s correlation analysis and path analysis were conducted to examine the relationships between and influences of taekwondo ability, GRIT, and academic achievement of university students who participated in a taekwondo class. In addition, all data used in this study were verified for normality by a normality test.

## 3. Results

### 3.1. Changes in GRIT

The differences in the changes in GRIT between participants and non-participants are presented in Table 1. The average of the participants’ GRIT before class participation was 35.77, and it has increased to 38.94 after class participation. The average of the non-participants’ GRIT was increased from 37.27 to 37.77 after a year, and there was a statistically significant difference (t = −2.138, *p* < 0.05). The results showed that the averages of both the experimental group and the control group increased, but that the amount of GRIT change of participants in the experimental group was higher.

### 3.2. Correlation Analysis

The correlations between participants’ taekwondo ability, GRIT, and academic achievement are listed in Table 2. The results showed that participants’ taekwondo ability had statistically significant correlations with GRIT (*r* = 0.418, *p* < 0.001) and academic achievement (*r* = 0.424, *p* < 0.001). They also showed that participants’ GRIT had a statistically significant correlation with academic achievement (*r* = 0.385, *p* < 0.001).

### 3.3. Path Analysis

Path analysis was performed to verify the correlation between participants’ taekwondo ability, GRIT, and academic achievement, and the results are presented in Table 3 and Figure 2. The results indicate that the *x^2^* (chi-Square) and degrees of freedom (*df*) of this study model were both 0 and that the *p*-value was 1.00, and it was accordingly judged as a perfect model.

According to Table 4, the higher the taekwondo ability and GRIT of participants, the higher the academic achievement, and the taekwondo ability and GRIT explain about 23.2% of the academic achievement. The results also indicated that the higher the taekwondo ability, the higher the GRIT, and that the taekwondo ability explains about 17.5% of GRIT.

According to Table 5, the participants’ taekwondo ability had a direct effect on their academic achievement, and the total effect of taekwondo ability for academic achievement increased to 0.424 (t = 6.654, *p* < 0.01) due to the indirect effect. In addition, GRIT had a direct effect on academic achievement; it had no indirect effect. Therefore, the total effect of GRIT for academic achievement was 0.252 (t = 3.709, *p* < 0.01).

The participants’ taekwondo ability had a direct effect on GRIT and no indirect effect. Therefore, the total effect of the taekwondo ability on GRIT was 0.418 (t = 6.540, *p* < 0.01).

## 4. Discussion

The results of this study supported both “hypothesis 1” and “hypothesis 2”. The GRIT of participants who participated in taekwondo class for a year was improved compared to their GRIT before participating in taekwondo class, and their rate of increase was larger than that for non-participants. These findings support previous studies reporting that exercise and sports activities improve GRIT [6,17], as well as Reed’s study reporting that people participating in exercise had higher GRIT strength [15].

Taekwondo is widely known throughout the world for its image of competition (sparring) with opponents wearing protective equipment in the Olympic games. However, this is only part of the reality of taekwondo which has been formalized into a sport. When taekwondo training begins, practitioners are given priority in learning about taekwondo theory and principles, such as history, philosophy, culture, and the spirit of taekwondo; the basic movements of taekwondo and practical skills such as poomsae and competition are trained in the process of this theoretical and principle-based education [22]. When the practitioner’s practical skills begin to grow, their taekwondo instructors apply psychological and emotional education, such as lessons in patience, sincerity, confidence, passion, stoicism, and having a never-give-up spirit to guide them to learn repeatedly while training. This systematic taekwondo curriculum [22,23,24] and on-site instruction helps practitioners improve not only their health, physical strength, and physical ability, but also their psychological, social, emotional, and spiritual aspects [25,26,27,28,29,30,31].

The students who participated in the study practiced taekwondo training for a year, and most of them passed the class. In addition to taekwondo training, the taekwondo class continuously provided feedback on step-by-step achievements based on educational guidance, such as “you can do it”, “repeat until you succeed”, “don’t give up”, “think positively”, “start over even if you fail”, “be tenacious”, “don’t be afraid to challenge yourself”, and “do your best”. Such content of the taekwondo class is similar to the learning contents and methods for GRIT growth [4,6,7,32]. After all, according to Duckworth’s research showing that GRIT is created by learning, not innate qualities [4,7], university students majoring in science and engineering who participated in the study are considered to have learned GRIT simultaneously through taekwondo class, and thereby showed the results of their GRIT growth.

The results of this study show that university students who did not participate in taekwondo class also showed slightly increased GRIT. In this study, the university classes taken by the non-participants were mostly the same as those taken by the participants, aside from the taekwondo class. They all resided in the same dormitory conditions, and their grades were also the same. Still, their personal tendencies, thoughts, hobbies, activities, and social life could not be controlled. University students are faced with new environments and situations when entering a university. In this process, they must adapt to new disciplines and people, and thus achieve personal development. While these uncontrolled parts are considered to be the result, it is worth noting that the growth rate of GRIT in participants is significantly higher than that in non-participants.

Next, the results of this study supported both “hypothesis 3” and “hypothesis 4”. The taekwondo ability of university students majoring in science and engineering who participated in taekwondo class for a year had an influence on their GRIT and academic achievement, and their GRIT also influenced academic achievement. The *x^2^* (chi-Square) and degrees of freedom (*df*) of this study model were both 0 and the *p*-value was 1.00, and it was accordingly judged as a perfect model [33,34,35]; the structural relationship between taekwondo ability, GRIT, and academic achievement was confirmed. These results indicate that it is possible to predict the GRIT results that can be achieved by participating in taekwondo class and developing taekwondo ability. This has important implications in terms of control over the future, because, generally, if studies focus on explaining social phenomena, the confirmation of structural relationships makes it possible to predict future results by clearly presenting the measurement errors as well as causal relationships [35], and predictable results can be controlled. Therefore, it can be concluded that for the growth of GRIT and academic achievement, it is better to participate in a taekwondo class and improve the taekwondo ability rather than making hypothetical choices not backed by evidence.

The statistical results of this study partially support the research of Greed, Cosgrove et al., and Kitano et al. [15,16,17], which reported on the relationship between GRIT of exercise and sports, and they also support the studies of both Mun and Kim, each of whom reported that taekwondo affects academic achievement and GPA [36,37]. They also support previous studies that reported on the relationships between and influences of GRIT and academic achievement [9,10,11,12,13,14].

Duckworth stated that growing GRIT requires “qualitatively different exercises” rather than those simply focusing on time and frequency of action [7]. Likewise, participating in taekwondo class does not guarantee that GRIT improvement or academic achievement will be achieved. The purpose of qualitative participation in taekwondo class is to enhance the taekwondo ability. Taekwondo ability refers to the ability to acquire not only practical skills but also general knowledge of taekwondo and to reflect practitioners’ attitudes and mindsets as taekwondoists in their lives [38]. Instructors in taekwondo class focus on having a positive impact on the practitioners’ behaviors and spirits through the training process rather than participating in the exercise itself [18]. This peculiar aspect of taekwondo education changes students’ thoughts and behaviors, and these changes are believed to have direct and indirect effects on taekwondo ability as well as GRIT and academic achievement. In summary, the main reason for the GRIT growth and academic achievement of university students majoring in science and engineering is not their participation in taekwondo class, but rather the taekwondo abilities they developed through participating in class; this growth in their taekwondo ability caused changes in the students’ thoughts and behaviors due to the external and internal values of taekwondo or the unique educational characteristics of taekwondo.

## 5. Recommendations and Conclusions

The purpose of this study is to confirm changes in the GRIT of participants and non-participants as well as the structural relationship between GRIT and academic performance by verifying the effects of taekwondo ability on GRIT and academic achievement, and of GRIT on academic achievement. The results indicated that the participants’ GRIT showed a statistically significant increase after a year, with a rate of increase that was higher than that of non-participants. It was also found that the taekwondo ability had significant correlations with and influences on GRIT and academic achievement, and that GRIT also had a significant correlation with and influence on academic achievement. In conclusion, taekwondo training for university students majoring in science and engineering improves their GRIT. In addition, the taekwondo ability has a positive effect on GRIT and academic achievement, and GRIT also has a positive effect on academic achievement, thereby forming a structural relationship between taekwondo ability, GRIT, and academic achievement.

Based on this study, the recommendations for subsequent studies are as follows.

First, in this study, we placed more importance on the development of taekwondo ability than on participation in taekwondo class. However, it is difficult to identify the causes and priorities that directly affect the formation of taekwondo ability in taekwondo class. Therefore, if the study in this part progresses in the future, it is believed that more efficient taekwondo classes can eventually be conducted.

Second, in this study, conclusions were drawn through a one-year educational experiment. If possible, it would be beneficial to compare the results of this study with a longitudinal study applied over a longer period.

Third, this study was conducted considering taekwondo. Beyond taekwondo, there are various martial arts and martial arts sports that have similarly positive impacts on humans in other countries. It is therefore considered necessary to examine whether these research results are consistent across different types of events.

## Figures and Tables

**Figure 1 ijerph-18-05361-f001:**
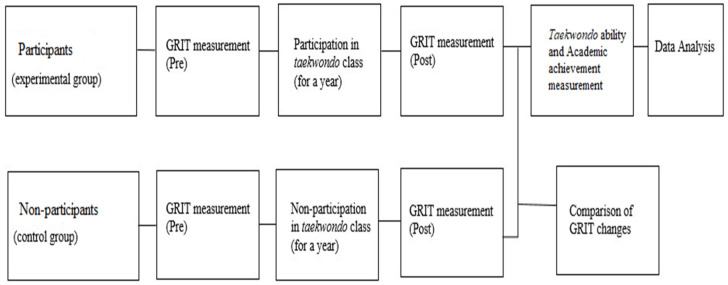
Research procedure.

**Figure 2 ijerph-18-05361-f002:**
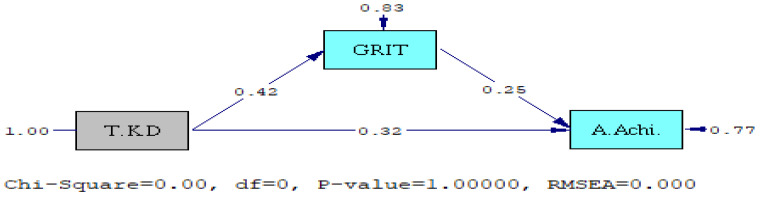
Path model.

**Table 1 ijerph-18-05361-t001:** Changes in GRIT of participants and non-participants.

	Students Who Participate in Taekwondo Class (*n* = 204)		Students Who Do Not Participate in Taekwondo Class (*n* = 101)	
	M	SD	M	SD
Pre	35.77	4.74	37.27	5.22
Post	38.94	5.13	37.77	5.03
t-value	−19.374		−2.138	
*p*-value	0.000		0.035	

**Table 2 ijerph-18-05361-t002:** Correlations of participants’ taekwondo ability, GRIT, and academic achievement *n* = 204.

Variable	1	2	3
Taekwondo Ability	1.00		
GRIT	0.418 ***	1.00	
Academic Achievement	0.424 ***	0.385 ***	1.00

*** *p* < 0.001.

**Table 3 ijerph-18-05361-t003:** Goodness of fit index of path analysis of taekwondo level, GRIT, and GPA.

	*x* ^2^	*df*	RMSEA
Research model	0.00	0	0.000

**Table 4 ijerph-18-05361-t004:** Parameter estimates of the research model.

Endogenous & Exogenous Latent Variable	Parameter Estimate (Standard Error)	Standard Solution	t-Value	SMC (*R^2^*)
Academic Achievement (*η*_2_)				0.232
GRIT (*η*_1_)	0.252 (0.068)	0.252	3.709 **	
Taekwondo ability (*ξ*_1_)	0.319 (0.068)	0.319	4.697 **	
GRIT (*η*_1_)				0.175
Taekwondo ability (*ξ*_1_)	0.418 (0.064)	0.418	6.540 **	

** |T| > 2.58 (*p* < 0.01).

**Table 5 ijerph-18-05361-t005:** Total and indirect effects of the research model.

Endogenous & Exogenous Latent Variable	Direct Effects	Indirect Effects	Total Effects	t-Value
Academic Achievement (*η*_2_)				
GRIT (*η*_1_)	0.252	-	0.252	3.709 **
Taekwondo ability (ξ_1_)	0.319	0.105	0.424	6.654 **
GRIT (*η*_1_)				
Taekwondo ability (*ξ*_1_)	0.418	-	0.418	6.540 **

** |T| > 2.58 (*p* < 0.01).

## Data Availability

The data presented in this study are available on request from the corresponding author. The data are not publicly available due to student privacy policy of the university that provided the research data.

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
