# Peer review of "The Relationship between Changes in GRIT, Taekwondo Ability, and Academic Achievement of University Students Majoring in Science and Engineering and Participating in Taekwondo Class"

_ijerph, 2021, doi:10.3390/ijerph18105361_

Round 1

Reviewer 1 Report

The study submitted focuses on the analysis of the effects of a taekwondo training program, based on a pre-post test design with a control group in first year university students. Despite the potential interest of the analysis, the following improvements are suggested, in order to achieve greater scientific rigor and impact:

Instroduction

Please deepen the justification of the relevance of the research problem by conducting a further review of the available (updated) international scientific literature.

Materials and Methods

Please reorganize this section considering international standards of scientific research: description of participants and type of sample (well written in subsection 2.1.); instrument (description of the instrument applied -GRIT scale-, whose adaptation needs more empirical evidence on its validity and reliability); design and procedure (included, in a segmented way, in section 2); data analysis (partially described in subsection 2.2.5.; although it is understood that Pearson correlations were applied, please include what type of correlation was carried out). Finally, include empirical evidence on the fulfillment of statistical assumptions for the application of parametric statistics (paired t-tests and Pearson correlations).

In relation to Path Analysis (extension of multiple regression analysis) -PA-, the assumptions related to sample size, independence of errors, uni- and multivariate normality, linearity, multicollinearity, recursion, interval level of measurement and acceptable levels of reliability should be evaluated. Most of the statistics used in the PA assume that the multivariate distribution is normal. Especially, a violation of this assumption is problematic because it could affect the accuracy of statistical tests, incorrectly suggesting that the model fits the data well or poorly.

Author Response

Thank you very much for the review of my study.

As suggested by the reviewer, I did my best to make corrections, but I think it is not enough. Any suggestions for further corrections would be appreciated.

Reviewer 2 Report

From my point of view, it is fascinating the publication. 

I am not an expert in this subject, so I do not know about the novelty. 

I have only two comments:

- Acronyms, such as GRIT, should not be included in the abstract. But its definition should stay in the abstract

- Line 22  orthographic mistake

Author Response

From my point of view, it is fascinating the publication.

I am not an expert in this subject, so I do not know about the novelty.

Thank you very much for the review of my study.

As suggested by the reviewer, I did my best to make corrections, but I think it is not enough. Any suggestions for further corrections would be appreciated.

I have only two comments:

- Acronyms, such as GRIT, should not be included in the abstract. But its definition should stay in the abstract

I couldn't find a substitute for GRIT. Although GRIT is an acronym, it acts like a proper noun. Instead, a description of GRIT is provided at the beginning of the abstract. If the reviewer suggests it, I will correct it.

- Line 22 orthographic mistake

I'm sorry. I don't know what is wrong for 'line 22'.

Line 22 : It was also found that taekwondo ability, GRIT, and academic achievement had significant correlations with each other.

Reviewer 3 Report

Dear authors, thank you for your work.
I list a number of areas for improvement for the publication of your manuscript.

1.- Introduction: It is difficult to justify theoretically what has been exposed in only one and a half pages. I recommend the authors to generously expand this part and introduce similar background information to provide preliminary data.

Method: This section is very well described and argued. I have nothing to add.

Results: I do not see any statistical error and the approach is correct. Only one question, was SPSS also used for the path model? On the other hand, I am not clear about the need to include table 3 and figure 2. If other reviewers do not say anything about it, they should be left, but if another reviewer, like me, sees that this information is unnecessary, the authors should consider removing it.

Discussion: It is well presented but I recommend that when the authors expand the introductory section, these new references should be used to qualify the discussion. In other words, revising the introduction implies a subsequent revision of the discussion.

Recommendations and conclusions: I recommend that authors merge the two sections and present the recommendations as limitations and prospective within the conclusion section. Furthermore, they should integrate practical and theoretical implications of the main findings of their study in the conclusion section.

Author Response

Dear authors, thank you for your work.

I list a number of areas for improvement for the publication of your manuscript.

Thank you very much for the review of my study.

As suggested by the reviewer, I did my best to make corrections, but I think it is not enough. Any suggestions for further corrections would be appreciated.

1.- Introduction: It is difficult to justify theoretically what has been exposed in only one and a half pages. I recommend the authors to generously expand this part and introduce similar background information to provide preliminary data.

Thank you very much for the review. I agree with your review.

However, in the international scientific literature, Taekwondo research focuses on training, coaching, rehabilitation, and injury, making it difficult to further review previous studies. I hope to seek the reviewer's consideration.

Method: This section is very well described and argued. I have nothing to add.

Thank you very much for the review.

Results: I do not see any statistical error and the approach is correct. Only one question, was SPSS also used for the path model? On the other hand, I am not clear about the need to include table 3 and figure 2. If other reviewers do not say anything about it, they should be left, but if another reviewer, like me, sees that this information is unnecessary, the authors should consider removing it.

In this study, SPSS was not used for path analysis, but LISREL was used. LISREL program recommends to present the results as shown in <Table 3> and <Figure 2>. I hope to seek the reviewer's consideration.

Discussion: It is well presented but I recommend that when the authors expand the introductory section, these new references should be used to qualify the discussion. In other words, revising the introduction implies a subsequent revision of the discussion.

Thank you very much for the review. I agree with your review.

However, in the international scientific literature, Taekwondo research focuses on training, coaching, rehabilitation, and injury, making it difficult to new references previous studies. I hope to seek the reviewer's consideration. Any detailed suggestions for further modifications would be appreciated.

Recommendations and conclusions: I recommend that authors merge the two sections and present the recommendations as limitations and prospective within the conclusion section. Furthermore, they should integrate practical and theoretical implications of the main findings of their study in the conclusion section.

Thank you very much for the review. I revised the sentence according to the reviewer's suggestion, and the revised content was marked in red in the text.